

# Lumbar extension peak-torque, muscle endurance and motor imagery in junior-elite basketballers with and without a history of low back pain: a pilot study

Tim Glazebrook[1,2], Jennifer Cooke[1], Gordon Waddington[1,3] and Sarah B. Wallwork[3,4]

[1] Australian Institute of Sport, Canberra, Australian Capital Territory, Australia
[2] SportsMed SA, Adelaide, South Australia, Australia
[3] Research Institute for Sport and Exercise, University of Canberra, Canberra, Australian Capital Territory, Australia
[4] IIMPACT in Health, University of South Australia, Adelaide, South Australia, Australia

## ABSTRACT

**Objectives.** Low back pain (LBP) is common in elite athletes. Several peripheral and central factors have been identified to be altered in non-athletic LBP populations, however whether these alterations also exist in elite athletes with LBP is unknown. The aim of this study was to determine whether elite basketballers with a history of persistent LBP perform worse than those without LBP at a lumbar muscle endurance task, a lumbar extension peak-torque task, and a lumbar motor imagery task.

**Method.** An observational pilot study. Twenty junior elite-level male basketballers with ($n = 11$) and without ($n = 9$) a history of persistent LBP were recruited. Athletes completed a lumbar extensor muscle endurance (Biering-Sorensen) task, two lumbar extensor peak-torque (modified Biering-Sorensen) tasks and two motor imagery (left/right lumbar and hand judgement) tasks across two sessions (48 hours apart). Performance in these tasks were compared between the groups with and without a history of LBP.

**Results.** Young athletes with a history of LBP had reduced lumbar extensor muscle endurance ($p < 0.001$), reduced lumbar extension peak-torque ($p < 0.001$), and were less accurate at the left/right lumbar judgement task ($p = 0.02$) but no less accurate at a left/right hand judgement task ($p = 0.59$), than athletes without a history of LBP. Response times for both left/right judgement tasks did not differ between groups (lumbar $p = 0.24$; hand $p = 0.58$).

**Conclusions.** Junior elite male basketballers with a history of LBP demonstrate reduced lumbar extensor muscle endurance and lumbar extension peak-torque and are less accurate at a left/right lumbar rotation judgement task, than those without LBP.

Corresponding author
Sarah B.
Wallwork,
sarah.wallwork@unisa.edu.au

## INTRODUCTION

Low back pain (LBP) is one of the most burdensome health conditions, affecting ~1 in five people at any given time (*Hoy et al., 2012*; *Vos et al., 2020*). For many, LBP lasts for the duration of tissue injury (~6 weeks) (*Wallwork et al., 2024*), but for 20–30% of people (*Traeger et al., 2016*) pain persists and becomes chronic. In athletes it is a similar story–LBP is thought to affect between one to six in 10 athletes at a given time (*Farahbakhsh et al., 2018*; *Trompeter, Fett & Platen, 2017*). In basketballers specifically, almost half of players (45.4%) are likely to report LBP at any given time (*Pasanen et al., 2016*). LBP is particularly evident amongst young males during periods of rapid growth and can be exacerbated with fluctuations in activity load (*Kujala et al., 1996*). Persistent LBP in athletes can be costly and lead to significant time away from training and competition (*Mortazavi, Zebardast & Mirzashahi, 2015*; *Olafsson et al., 2018*).

In non-athletic populations, people with persistent LBP demonstrate impairments to the motor system that manifest both in the periphery and within the central nervous system. For example, LBP is associated with localised lumbar muscle deconditioning (*i.e.,* poor endurance, strength, increased fatigability) (*Latimer et al., 1999*; *Steele, Bruce-Low & Smith, 2014*; *Süüden et al., 2008*), and as a result, many interventions for LBP now include strengthening the lumbar and trunk muscles using individual exercise programs or engaging in group exercise classes such as Pilates (*Clael et al., 2021*). Such interventions that target the periphery (*e.g.,* exercise therapy) have had limited success for reducing LBP (*Hayden et al., 2021*). Impairments also exist in the central nervous system in people with persistent LBP (*Wand et al., 2011*). For example, the tactile and cortical proprioceptive representations (*i.e.,* the brain-grounded neural representations) are thought to be disrupted in people with persistent LBP (*Bowering et al., 2014*; *Bray & Moseley, 2011*; *Moseley, 2008*). People with persistent LBP perform worse than healthy controls on: a motor imagery task that requires participants to judge whether a pictured model has their trunk rotated to the left or right (*i.e.,* a left/right lumbar rotation judgement task) (*Bowering et al., 2014*); a touch discrimination task to the back that involves identifying the shortest distance between two points at which two points can be detected, instead of one (*i.e.,* tactile acuity) (*Luomajoki & Moseley, 2011*); a cognitive task that involves tracing the outline of one's back as it is perceived 'in the mind's eye' (*Moseley, 2008*). Neuroimaging studies show evidence of both structural and functional brain changes in people with persistent LBP, that are consistent with these functional disruptions (*Wand et al., 2011*). Furthermore, treatments aimed at normalising these representations have demonstrated some promise in several pain conditions (*Bagg et al., 2022*; *Bowering et al., 2013*; *Moseley, Zalucki & Wiech, 2008*).

It is unclear whether the same peripheral and central impairments seen in the general population also exist in athletes with LBP. Male collegiate athletes with non-specific LBP have been shown to have significantly lower core muscle endurance than pain-free athletes (*Abdelraouf & Abdel-Aziem, 2016*). Furthermore, impairments in spatial processing of tactile and auditory data (*i.e.,* less able to locate a tactile or auditory stimulus) exist in athletes with persistent posterior thigh pain (*Summers et al., 2020*). These findings in athletic LBP populations are certainly consistent with findings in the general LBP
population, but further enquiry in a range of athletic populations is needed. If similar impairments extend into athletes with LBP, therapeutic approaches that target these impairments may also be indicated for athletes with persistent LBP (*Wallwork et al., 2016*).

Therefore, the primary aim of this study was to investigate whether elite-level athletes with a history of persistent LBP show evidence of both peripheral and centrally oriented impairments that are commonly observed in non-athletic populations. Specifically, this study sought to establish whether a junior elite basketball cohort with a history of persistent LBP perform worse than those without LBP at a lumbar muscle endurance task (Biering-Sorensen test), a lumbar extension peak-torque task (modified Biering-Sorensen test), and a motor imagery task (left/right lumbar rotation judgement task). It was hypothesized that athletes with LBP would perform worse than pain-free controls. The secondary aims were related to clinical utility: to assess the (a) inter-assessor agreement of the modified Biering-Sorensen test (measure of lumbar peak-torque), and to (b) to assess the relationship between lumbar extensor muscle endurance and lumbar peak-torque. The latter was included as a secondary aim as a strong relationship between lumbar extensor muscle endurance and lumbar peak-torque may have the potential to expediate screening procedures for clinicians working with elite-level athletes.

## MATERIALS & METHODS

Twenty-six junior-elite male basketballers were recruited in Canberra, Australia. Athletes were eligible for the study if they played competitive basketball at the junior-elite level (*i.e.,* played at the national and/or international level) and had unrestricted training in the two weeks prior to testing. Athletes were excluded if they were experiencing an episode of LBP at the time of testing or within the two weeks prior to testing. Engaging elite athletes for involvement in health-related research has previously been described as presenting significant challenges to recruitment and subsequent methodologic issues in these 'hard to recruit' populations (*Edouard, Branco & Alonso, 2014*). This is further detailed by *Schubring et al. (2019)* who describe challenges including access, participation, and anonymity issues for athletes participating in research studies. Therefore, we utilised a sample of convenience that was the entire squad of athletes at the junior-elite level available at the training institute site where the research team were located. No sample size calculation was performed. Due to the small sample size, this study was continued as a pilot study. Ethical approval was gained from the Australian Institute of Sport Ethics Committee (#20190405) and all athletes provided written informed consent prior to participation. For participants under the age of 18 years, their parent or legal guardian also provided written informed consent.

Lumbar extension muscle endurance was assessed using the Biering-Sorensen test. The Biering-Sorensen test is a reliable and valid assessment of the isometric endurance of the trunk extensor musculature and has been shown to be able to discriminate between subjects with and without nonspecific LBP (*Biering-Sorensen, 1984*; *Latimer et al., 1999*). Athletes were positioned prone on a treatment plinth with the lower half of their body (below the level of the anterior superior iliac spine) strapped to the couch at three positions: at the ankles (close to malleoli), at the knee creases, and at the level of the greater trochanter of

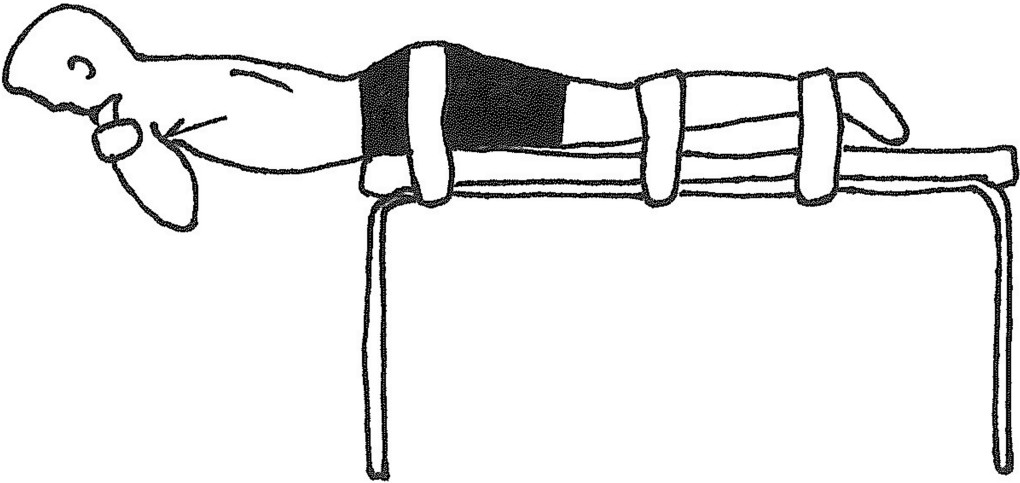

**Figure 1** Testing position for the Biering-Sorensen task and the modified Biering-Sorensen task.

the femur (see Fig. 1). Straps were tightened as firmly as possible within the athletes' level of comfort. Prior to the test commencing, the athlete could rest their trunk on a chair. The athlete was instructed to lift their upper trunk clear of the chair, whilst placing the arms across the chest and maintain the trunk in neutral alignment for as long as possible. They were told to stop the test if they experienced any pain or discomfort. No athletes reported pain or discomfort during testing and were able to complete all tests. The experimenter measured the time in seconds that the athlete could remain in this position.

Lumbar peak-torque was measured using a variation of the Biering-Sorensen task based on work previously performed in assessment of muscular peak-torque wherein testing through hand-held dynamometry was established as being reliable despite some intertester bias based on tester characteristics (*Thorborg et al., 2013*). Athletes completed this task twice with two separate assessors (trained physiotherapists) to enable inter-rater reliability assessment, with at least 15 min between tests (greater than the recommended 3–5 min rest interval) (*Freitas De Salles et al., 2009*). To complete the test, athletes were positioned as per the Biering-Sorensen test as described above (see Fig. 1). Athletes were instructed to lift their upper trunk, with their hands across their chest, and maintain the trunk in a neutral position. The experimenter placed a calibrated electronic hand dynamometer perpendicular to the spine, at the level of the spine of the scapulae, and applied a force sufficient to 'break' their neutral position. Instructions of "go ahead, push, push, push, push and relax" were given and vigorous verbal encouragement was provided.

The integrity of the cortical proprioceptive representation was assessed using an implicit motor imagery task, the most well-established of which is a left/right body judgement task. The left/right body judgement task involves making a judgement about whether a presented image of a body-part belongs to the left or right side of the body (or is rotated to the left or right side of the body) (*Parsons, 1987*). Performance is measured using two metrics: response time (RT) and accuracy. A longer RT is thought to demonstrate disruptions to spatial processing of the body part, whereas reduced performance is thought

to demonstrate altered cortical proprioceptive maps (*i.e.,* the brain-grounded neural representations that are used to plan and execute movements) of the affected body part (*Bray & Moseley, 2011*). The task was undertaken on an iPad using the Recognise software (http://www.noigroup.com, Adelaide, Australia). This approach has demonstrated good to excellent concurrent validity with the desktop version of the task, as well as good to excellent test-retest reliability (*Williams et al., 2019*).

Athletes completed two left/right judgement trials: the first was a left/right judgement of lumbar rotations (*i.e.,* judging whether the trunk of the person in the image was rotated or laterally flexed to their left or their right side) and the second was a left/right hand judgement task (*i.e.,* judging whether the image presented was a left hand or a right hand). The hand judgement task was included as a control condition, whereby if between group differences were found for the back task, but not the hand task, that would indicate body-part specific disruptions to the cortical proprioceptive representation, whereas if between group differences were found for both tasks, that would indicate a disruption to spatial processing more broadly. Both trials included 40 images of equal left and right sided images, and both tasks were preceded by a practice trial of back images and hand images, respectively.

Athletes attended the Australian Institute of Sport (AIS) Physiotherapy clinic on two occasions, two days apart (>48 hrs). Athletes were asked to continue with their normal training activities in the days prior to and between testing and encouraged to report any concerns regarding their ability to continue through the study. Each session took a total of 20 min and was comprised of familiarisation with the testing procedure through written and verbal communication, followed by formal assessment. Testing sessions were scheduled across days involving fewer or less rigorous training sessions and out of competition periods.

Prior to Testing Day 1, athletes were briefed and given a copy of the patient information and consent form describing the activities asked of them and allowed a 24hr perusal. Those agreeing to participate were asked to submit completed consent forms before any further involvement in the study.

On Testing Day 1, athletes were asked to complete two screening questionnaires (Oswestry LBP Disability Questionnaire (OLBPDQ) and the Physical Activity Readiness Questionnaire (PARQ)) to determine immediate LBP status and therefore suitability to continue with the testing protocol. Athletes reporting current LBP were excluded from further participation in the study. Athletes deemed appropriate for inclusion were asked to perform the modified Biering-Sorensen test. Prior to completing the task, two non-maximal Biering-Sorensen practice trials were performed to ensure correct test positioning. Each practice trial was 5 s in duration followed by 10 s rest. Participants were given 20 s to rest between the final practice trial and the maximal test effort. On completion of the test, athletes undertook the left/right judgement tasks on the iPad, before returning for their second modified Biering-Sorensen test, with at least fifteen minutes between the two modified Biering-Sorensen tests to allow for sufficient recovery.

On Testing Day 2, athletes were asked to re-evaluate their current pain state by completing the questionnaires a second time. Athletes reporting current or previous

LBP were interviewed to gain a brief history of the pain and current symptoms and were withdrawn from further assessments of strength if they were thought to be at risk of further exacerbation. If deemed appropriate for continued involvement, athletes underwent the full Biering-Sorensen test.

Data were analysed using IBM SPSS Statistics for Windows, Version 27.0, Armonk, NY, IBM Corp. To determine whether athletes with a history of persistent LBP performed worse than those without LBP, independent samples t-tests were undertaken between groups for peak-torque (for both Assessor 1 and 2), muscle endurance, RT and accuracy for judgements of lumbar rotations, and RT and accuracy for judgements of hands. Brown-Forsythe tests for equal variance and Shapiro–Wilk tests of normality were undertaken. Where tests of normality were violated, the non-parametric Mann–Whitney U test was used. To assess inter-rater reliability of the modified Biering-Sorensen test, an intra-class correlation coefficient (Model: two-way random effects; Definition: 'consistency') was computed based on the recommendations of *Koo & Li (2016)* and a Bland-Altman plot was constructed to check for systemic bias, variability and agreement. Correlations were considered poor, moderate, good and excellent if the ICCs were less than 0.5, between 0.5 and 0.75, between 0.75 and 0.9 and greater than 0.9, respectively. Agreement was considered sufficient if the mean difference between the two assessors' scores was close to zero (zero was within the 95% CI). To assess the relationship between lumbar extensor muscle endurance and lumbar peak-torque, intraclass correlation coefficients were calculated between the two measures of lumbar peak-torque (Assessors 1 and 2) and lumbar muscle endurance. Significance was set at $\alpha = 0.05$.

## RESULTS

Twenty-six athletes were recruited in this pilot study. Four athletes were excluded due to pain within the previous 2 weeks and two athletes withdrew due to logistics associated with their basketball program. Twenty athletes completed the full study procedure (age (mean $\pm$ SD): 17.24 $\pm$ 0.82 years). Of those 20, 11 athletes reported a history of LBP.

Athletes with a history of LBP performed worse than those without a history of LBP at the lumbar muscle endurance task (Tests for Equal Variance (Brown-Forsythe, $p = 0.12$) and Normality (Shapiro–Wilk, $p = 0.49$) of the data were passed, t(18) = 6.26, $p < 0.001$; mean difference 54.94), and the lumbar extension peak-torque task (Assessor 1: Mann–Whitney U test: $p = 0.001$; Assessor 2: Tests for Equal Variance (Brown-Forsythe, $p = 0.49$) and Normality (Shapiro–Wilk, $p = 0.25$) of the data were passed, t(18) =5.43, $p < 0.001$, mean difference 90.37). There was excellent correlation between the two physiotherapy assessors for measures of lumbar extension peak-torque, when participants with a history of low back pain and those without low back pain were assessed as one combined group (ICC 0.979, 95% CI [0.947–0.992]), although the overall agreement between the two assessors' scores was poor (mean difference: 86.55, 95% CI [49.89–123.21]) (see Fig. 2). The degree of correlation between the two assessors was also determined separately for those with a history of low back pain ($n = 11$, ICC 0.189, 95% CI [0.099–0.491]) and for participants without a history of low back pain ($n = 9$, ICC 0.364, 95% CI [0.139–0.791]) (see Fig. 2). There was good correlation between measures of lumbar muscle endurance and lumbar

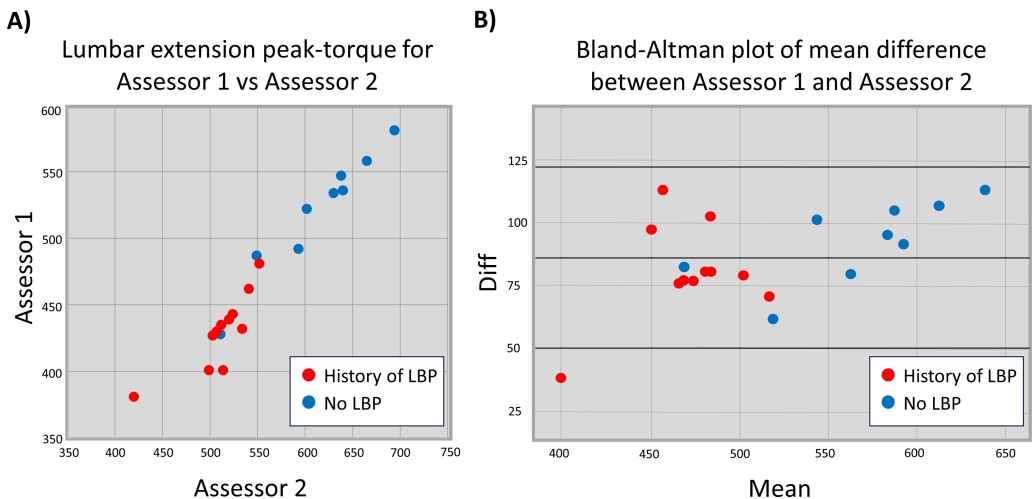

**Figure 2 Assessor comparison for lumbar extension peak-torque.** (A) Scatterplot showing values from Assessor 1 and Assessor 2 for lumbar extension peak-torque, and (B) a Bland-Altman plot showing the mean difference between the two assessors (middle line) and the 95% confidence intervals. Note that zero does not fall within the 95% CI, demonstrating poor agreement between the two assessors.

extension peak-torque (Assessor 1: ICC 0.752, 95% CI [0.473–0.894]; Assessor 2: ICC 0.820, 95% CI [0.599–0.924]).

Athletes with a history of LBP were less accurate than those without a history of LBP at the left/right lumbar discrimination task (Tests for Equal Variance (Brown-Forsythe, $p = 0.82$) and Normality (Shapiro–Wilk, $p = 0.22$) of the data were passed, t(18) =2.56, $p = 0.02$, mean difference 6.57%, 95% CI 1.17% −11.96%), but not at the left/right hand judgement task (Tests for Equal Variance (Brown-Forsythe, $p = 0.49$) and Normality (Shapiro–Wilk, $p = 0.25$) of the data were passed, t(18) =.548, $p = 0.59$, mean difference 2.53%, 95% CI −7.15% −12.21%). RT for both left/right discrimination tasks did not differ between groups (lumbar: Tests for Equal Variance (Brown-Forsythe, $p = 0.07$) and Normality (Shapiro–Wilk, $p = 0.67$) of the data were passed, t(18) = 1.32, $p = 0.21$, mean difference 0.10 ms, 95% CI [−.07 ms–0.26 ms]; hand: (Tests for Equal Variance (Brown-Forsythe, $p = 0.80$) and Normality (Shapiro–Wilk, $p = 0.44$) of the data were passed, t(18) = 0.1, $p = 0.93$, mean difference 0.02 ms, 95% CI [−0.45 ms–0.49 ms]) (see Fig. 3).

## DISCUSSION

This pilot study investigated whether male junior elite-level athletes with a history of LBP show evidence of the same peripheral and central impairments that are observed in non-athletic populations. The results showed that athletes with a history of LBP performed worse than athletes without a history of LBP at a lumbar muscle endurance task, a lumbar peak-torque task, and lumbar motor imagery task. These findings are aligned with findings in non-athletic populations (*Bowering et al., 2014*; *Bray & Moseley, 2011*; *Shaw et al., 2024*).

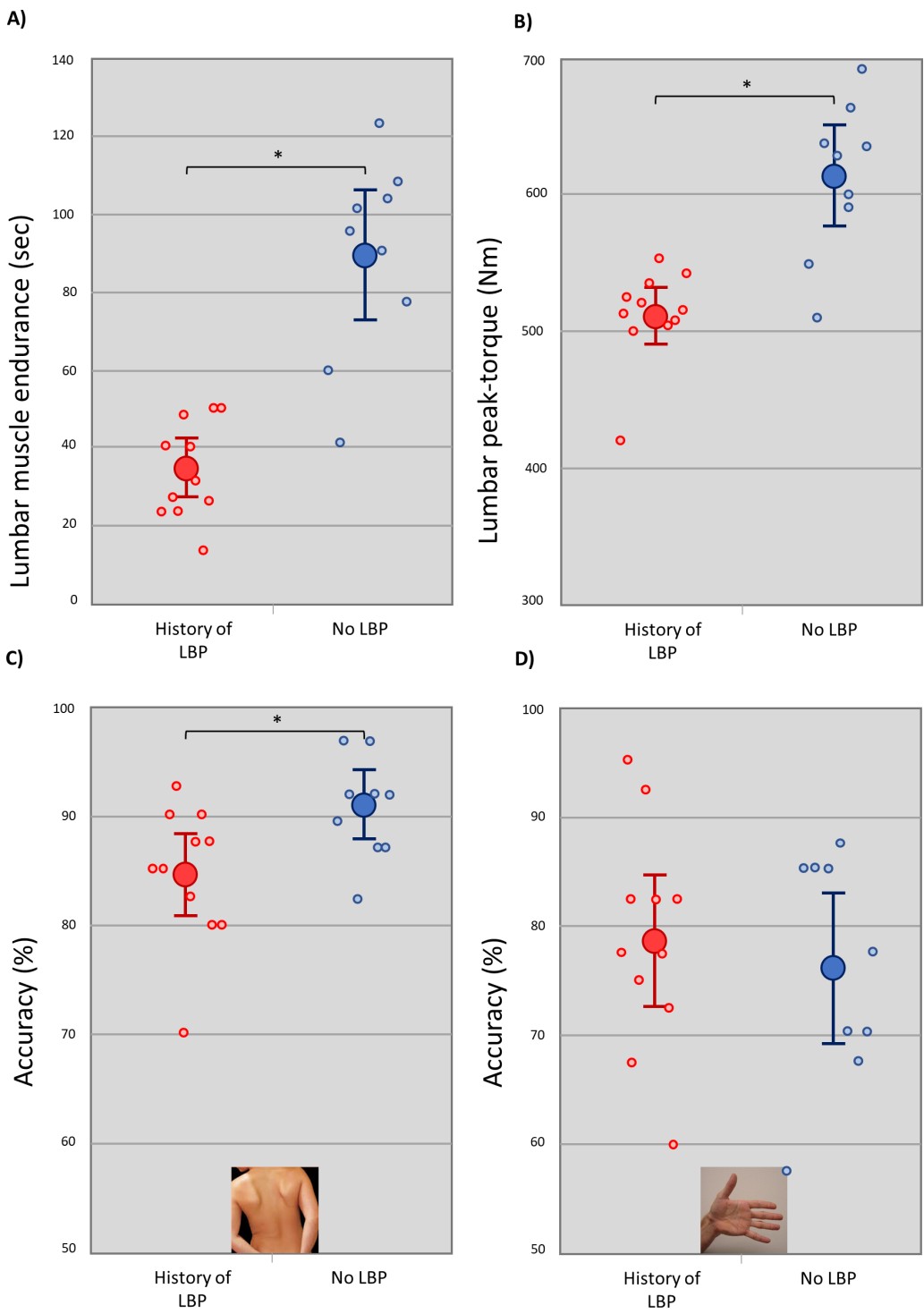

**Figure 3** **Left/right judgement performance between athletes with and without a history of low back pain.** Between group comparisons (mean (large circles) and 95% confidence intervals) for (A) lumbar muscle endurance, (B) lumbar peak-torque, (C) left/right lumbar judgement accuracy, and (D) left/right hand judgement accuracy. Small circles represent individual athletes. An asterisk (*) indicates significance at 0.05.

Athletes with a history of LBP performed worse at both the lumbar endurance and the lumbar peak-torque tasks than athletes without a history of LBP and there was good correlation between measures for lumbar muscle endurance and lumbar peak-torque across our athlete population. While reduced lumbar muscle endurance in people with LBP is well established in the literature (*Steele, Bruce-Low & Smith, 2014*), the finding of reduced lumbar peak-torque is less established and suggests impaired lumbar muscle function more broadly in athletes with LBP. The use of peak torque as a screening modality for groin-related pain (*O'Connor, 2004*), hamstring pain (*Wollin, Purdam & Drew, 2016*) and shoulder pain (*Soderberg & Blaschak, 1987*) is well recognised and implemented in clinical practice, and as such, further investigation into peak-torque measures as a screening modality for LBP may be indicated. The findings of the current study are consistent with those in non-athletic populations (*Latimer et al., 1999*), and raises the possibility that treatments that are directed to peripheral mechanisms in non-athletic populations, such as increasing muscle strength and endurance, might also be translatable into athletic populations. This line of enquiry remains to be tested.

Athletes with a history of LBP were less accurate at the left/right lumbar judgement task, but not the left/right hand judgement task, than those without a history of LBP. Finding deficiencies in the lumbar task, but not the hand task, is suggestive of a body-part specific disruption to the cortical proprioceptive maps (*Bray & Moseley, 2011*). These results are not surprising and are consistent with several imaging studies that show cortical brain changes (gray matter) to the somatosensory, motor, and prefrontal cortices in people with low back pain (*Elgueta-Cancino, Schabrun & Hodges, 2018*; *Flor et al., 1997*; *Medrano-Escalada et al., 2022*; *Ung et al., 2012*). While it is unclear why or how these changes may occur, there is some evidence to suggest that motor cortex reorganization may be associated with deficits in postural motor control in people with recurrent LBP (*Tsao, MP & Hodges, 2008*). The results of the current study are also consistent with findings from several pain conditions and populations including people with LBP (*Bowering et al., 2014*), knee osteoarthritis (*Stanton et al., 2012*), neck pain (*Wallwork et al., 2020*) and complex regional pain syndrome (*Moseley, 2004*). That no difference was found in RT between groups for either the hand or lumbar tasks, suggests against a spatially-defined disruption, which has been identified in people with chronic pathological limb pain (*Moseley, 2004*).

Reduced accuracy in the left/right judgement task has been suggested to reflect a disruption to cortical proprioceptive representations in the brain (*Bray & Moseley, 2011*). There are important considerations for this suggestion however: proprioceptive representations cannot, thus far, be quantified using neuroimaging in the same way that, for example, tactile (primary sensory cortex) and motor (primary motor cortex) can be. The neural substrates of left/right judgements, and the change in those substrates with training, have been elucidated to some extent (*Strauss et al., 2021*), but definitive statements about the neurophysiology underpinning disrupted left/right judgement performance, remain problematic. Notwithstanding, there is building evidence that so-called neural representation training (*Wallwork et al., 2016*; *Wallwork, Bellan & Moseley, 2017*) (*e.g.*, graded motor imagery (*Moseley et al., 2012*)—which incorporates left/right discrimination training) has been shown to reduce pain in both complex regional pain syndrome and

phantom limb pain (*Bowering et al., 2013*). Furthermore, a recent randomized clinical trial testing a Graded Sensorimotor Retraining intervention (which also includes left/right discrimination training) found small improvements in pain intensity in people with chronic low back pain (*Bagg et al., 2022*). Whether neural representation training might offer benefit for athletic populations with LBP (or other types of chronic pain) seems worthy of further investigation.

The secondary aims, which targeted clinical utility, also revealed potentially important gaps in knowledge. There was excellent correlation, but poor agreement, between assessors for measures of peak-torque. That is, while measures of peak-torque were well correlated, one assessor obtained consistently lower measures than the other assessor (mean difference: 86.55 Nm). This may be a reflection of individual assessors' abilities to produce force in 'breaking' the participant's hold in the test position. The two assessors included in this study had very different body-types, which may have impacted their ability to impart force on the participants (*Thorborg et al., 2013*). Strategies need to be considered to improve agreement between assessors, such as external fixation, which is less dependent on individual assessor body-types and strength.

This study had limitations. A very specific cohort of young elite-level male basketballers was included and therefore the results may not generalise beyond that cohort. Although research on strength and endurance is common in youth athletic populations, research on left/right judgements in that group is not. A larger sample size would also bring more confidence in our findings. This study should be considered a pilot study on which more research can be based. Detailed information on athlete demographics were not collected, including race, ethnicity, education, or a detailed history of their low back pain, limiting the ability to generalise our findings to other athlete populations. The order of assessments was not randomised and the potential effect of one on the other can therefore not be excluded. Specifically, the effect of fatigue on the endurance and peak force tasks, as well as the potential learning biases across all tasks. The lack of randomisation may have also been an influencing factor in the observed differences between the two assessors. Having said this, athletes were given sufficient time for recovery between the two maximal assessments; at least 15-minutes, exceeding the recommended rest period of 3–5-minutes (*Freitas De Salles et al., 2009*). Assessments were run across two days as a way of reducing potential follow-on effects. Finally, the protocol was not lodged prior to undertaking this study, something which is recommended in pain research (*Lee et al., 2018*).

## CONCLUSIONS

This pilot study provides further evidence that may indicate peripheral and central impairments in athletes with a history of LBP. This was evidenced by athletes with a history of LBP, but without symptoms in the previous two weeks or during testing, demonstrating reduced lumbar muscle endurance, reduced lumbar muscle peak-torque, and reduced accuracy at a lumbar left/right judgement task, then athletes without a history of LBP. The findings of this study align with what has been found in non-athletes with LBP. Further work investigating whether treatments that have shown some promise for reducing pain in non-athletic populations are also helpful for athlete populations, is warranted.

## ACKNOWLEDGEMENTS

We thank Professor Lorimer Moseley for helpful comments on earlier versions of this manuscript.

### Funding

Sarah B Wallwork is supported by a Leadership Investigator Grant awarded to Professor Lorimer Moseley, from the National Health & Medical Research Council of Australia ID1178444. The funders had no role in study design, data collection and analysis, decision to publish, or preparation of the manuscript.

### Grant Disclosures

The following grant information was disclosed by the authors:
The National Health & Medical Research Council of Australia ID1178444.

### Competing Interests

Tim Glazebrook works as a physiotherapist at SportsMed SA where he treats patients with musculoskeletal and sports injuries and pain. Sarah B. Wallwork receives payments for lectures on pain and rehabilitation.

### Author Contributions

- Tim Glazebrook conceived and designed the experiments, performed the experiments, analyzed the data, prepared figures and/or tables, authored or reviewed drafts of the article, and approved the final draft.
- Jennifer Cooke conceived and designed the experiments, performed the experiments, authored or reviewed drafts of the article, and approved the final draft.
- Gordon Waddington conceived and designed the experiments, analyzed the data, authored or reviewed drafts of the article, and approved the final draft.
- Sarah B. Wallwork conceived and designed the experiments, analyzed the data, prepared figures and/or tables, authored or reviewed drafts of the article, and approved the final draft.

### Human Ethics

The following information was supplied relating to ethical approvals (i.e., approving body and any reference numbers):
Australian Institute of Sport Ethics Committee.

### Data Availability

The raw data is available in the Supplementary File.

### Supplemental Information

Supplemental information for this article can be found online at http://dx.doi.org/10.7717/peerj.17508#supplemental-information.

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
