# Peer review of "Lumbar extension peak-torque, muscle endurance and motor imagery in junior-elite basketballers with and without a history of low back pain: a pilot study"

_PeerJ, doi:10.7717/peerj.17508_

## Round 0.1 · original submission · Major Revisions

Dear Authors,

Please revise the manuscript considering the reviewers' comments and suggestions.

Thank you.

Best regards.

Reviewer 1 ·

Basic reporting

Some sentences should be revised to avoid using the "we". The use of these expressions is especially evident in discussion section.
Line 163, a final stop is missing.
In the lines 58 and 59 ("Persistent LBP in athletes can be costly and lead to significant time away from training and competition.") a literature reference is missing. The same in lines 65 and 66. Again in 242-243.
In lines 248-249, the authors say that it is well established in the literature, but they only cite one reference from 1999 which does not correspond to a systematic review. I suggest being careful with the use of words or putting more references to support the claim.

Experimental design

The research appears to be within the aims and scope of the journal
Although not being a groundbreaking research, it contributes to the growth of science.


The research was carried out rigorously and presented in a structured way.
The authors reported that the lumbar extensors peak torque assessment was carried out by two different assessors. It wasn't clear to me which value was used for the comparison between the groups, whether the average of the two assessors, or whether only the data from one of the assessors was used.
The authors said that the persons were told to stop the test if they experienced any pain or discomfort. I recommend mentioning if there have been cases where this has happened, as it may influence the results.
The authors mentioned the differences between evaluators in their response to the peak torque, as well as mentioning in the limitations that the order was not randomized.
Could this factor also explain the differences in the strength of the lumbar extensors?
On lines 142-143, you mention 20 seconds of rest between training and assessment. How did you define this time? Was it through the pilot study? If the training repetitions reached high intensities, these 20 seconds may not have been enough to fully recover.
In line 157, they state that validity and repeatability has already been established. It is important to clarify that this reference points to the validity of the APP in relation to the desktop application, and not a validity in relation to what it is supposed to measure. Check if there are validity values for the desktop version or clarify that the validity is in relation to the desktop version.
Line 208, "Significance was set at p = 0.05." instead of "p" it should be alfa or establish the confidence level as 95%.
lines 220 to 235 could be clearer. Consider talking about differences between evaluators rather than agreement (same in the results section).
The type of ICC used must be explained in the paper. Consider using the ICC of absolute agreement since it takes systematic error into account.
It is not usual to report the values of the shapiro-wilk test.
The authors tested the correlation between lumbar muscle endurance and lumbar extension peak-torque. However, the objective is not clear in the introduction section.
In the discussion section, sometimes the text is a bit repetitive, focusing on the results observed, rather than on the discussion (240-242 and 245-246). I'd suggest talking about the results when you're discussing them. Otherwise, it's just a paragraph about results.
Lines 260 to 268 should explore potential reasons to explain the results. Consider whether the type of population could be a factor.
Lines 281-283, you said "Graded Sensorimotor Retraining intervention (which also includes left/right discrimination training) reduced pain in people with chronic low back pain (Bagg et al., 2022).". The authors of the article warn of the low effects of this intervention. This is one of the cases where a statistically significant difference may not be a clinically relevant difference. The authors took this precaution, and so should the authors who reference them.
Have you calculated the sample size? This is because they mention sample size as a limitation. You can always do a posteriori analysis and make the research more robust.

Validity of the findings

The authors chose a population with a history of pain, but without pain at the time of the assessment.
Although this fact is clear on the methodology, throughout the paper the group is referred to as the low back pain group, which may mislead some readers. I recommend that you should clarify this point, especially in the conclusion, where it says that the study provides evidence for people with persistent pain and after talks about people with a history of pain. In the conclusion, the population should be better clarified (people with a history of low back pain, but no symptoms in the last two weeks, and no symptoms during the evaluation).
I recommend that in the conclusion, line 309, you use an expression that allows for some degree of doubt, such as "may indicate". The outcomes evaluated do not fully represent the concepts of peripheral and central impairments

Reviewer 2 ·

Basic reporting

The manuscript presents a good quality writing. The introduction is appropriate and presents clear statements of their aims, along with a proper justification for conducting the study.

Experimental design

The current study presents several methodological limitations that could influence null-hypothesis significance testing (HNST) in the study:

1) No sample size calculations have been conducted. It is of great importance, stablishing alfa, and beta error, based on a previous effect size calculation (Cohens' d) for previously published works or based on a pilot sample. Having these data stated based on researcher's criteria and other author's recommendations, we will have greater certainty that the HNST will be conducted properly and with a controlled amount of bias.

In addition, these calculations must be conducted for your main outcome measure of interest, or instead, for all the included outcome measures (this would be highly recommended). This also applies for inter-rater reliability outcome measures (which usually require even greater samples).

2) It would be of great interest having conducted a randomization of the order of tasks, therefore, the possible fatigue / learning biases would have been reduced. This concerns, applies in specific for both endurance, and peak force tasks (randomizing the day they would conduct each task), and the order of left/right hand, and back laterality judgement tasks.

There are also limitations for characterizing the sample included in the study. It would have been of great interest, having knowledge of their age, sex, BMI, and other outcome measures (history of pain), usual pain intensity, etc. Central tendency (mean, median...), and variability parameters (SD, range, etc.), would have been appropriate to have them reported.

Validity of the findings

Clear limitations rise regarding the validity of the results of the present work. Sample size calculations should be highly recommended to be conducted, along with a proper randomization of tasks. The validity of these findings will respond clearer to authors' hypotheses.

---

## Round 0.2 · Minor Revisions

Dear authors,

Please revise the manuscript considering the feedback from reviewers 1 and 2.

Thank you.

Best regards,

Reviewer 1 ·

Basic reporting

The authors responded and improved the final document according to the comments made.

Experimental design

In general, the authors have responded adequately to the comments made. Below are some additional points:

Regarding the response to : "7. The authors mentioned the differences between evaluators in their response to the peak torque, as well as mentioning in the limitations that the order was not randomized. Could this factor also explain the differences in the strength of the lumbar extensors?", It could be interesting to include points from the response to the reviewer in the document. In particular, mentioning the 15 minutes of rest between tests (optional).

Regarding the response to : "8. On lines 142-143, you mention 20 seconds of rest between training and assessment. How did you define this time? Was it through the pilot study? If the training repetitions reached high intensities, these 20 seconds may not have been enough to fully recover."
Consider use the expression: "two non-maximal Biering-Sorensen practice trials" to be more clear.

Regarding the response to : "18. Have you calculated the sample size? This is because they mention sample size as a limitation. You can always do a posteriori analysis and make the research more robust.".
I understand the authors answer, however I recomend to explore power achieved with the effect size observed for the major objetctive or to all objectives, since there is a possibility of not having a sample size limitation.

Validity of the findings

The authors responded and improved the final document according to the comments made.

Additional comments

No adicional comments

Reviewer 2 ·

Basic reporting

The reviewed manuscript seems to not have properly adressed the comments suggested in the first review.

Experimental design

Authors seemed to claim the challenges and difficulty for recruiting elite athelete populations. The sample size is a key factor for reaching consistent findings that can contrast your hypotheses. Nevertheless, authors could have conducted a review of the prevalence of these subjects world-wide, or in their specific region (Australia), and base the inclusion of subjects on this prevalence, as it is usually done with rare diseases (although, in my opinion, it is not the same case).

The sample of pilot studies can also be justified with previous literature. This has not been pointed out in the manuscript. Several recommendations in the literature have been stated for pilot sample sizes, specially for randomized controlled trials.

No methodological modifications have been conducted concerning the randomization, or other processes for controlling the amount of learning/fatigue bias derived from the repeated assessments.

Another point to adress, concerns whether the inter-rater reliability measure (ICC) was aimed to be determined only from healthy junior-elite basketballers, only from LBP patients, or from both. Inter-rater realiability can vary depending on the consistency of the selected population. In the case of including LBP patients, as it seems to be the case, there is a relevant point to methodologically adress. LBP patients can present greater variability in lumbar peak force, derived from the presence of pain. Additionally, asssessors present biases during the evaluation process. This could also have been controlled randomizing the order in which evaluators assessed peak for in the first or second place.

In line with the comments of Reviewer nº1, I also highly recommend the use of Test of Homogeneity of Variances (for example, Levene's Test), as part of the verification process for employing the T-Test, in combination with the assessment of Normal distribution (Shapiro-Wilk test).

Validity of the findings

I highly recommend authors to conduct the previously mentioned suggestions from the first review process, with the ones suggested in the present review. There are important methodological procedures that must be adressed, so that the article exposes useful, consistent and reliable results.

---

## Round 0.3 · Minor Revisions

Dear Authors,

Reviewer 2 raised some concerns regarding the study, namely "Firstly, the methodology presented in the manuscript lacks the necessary quality expected for publication in PeerJ. Upon careful review, it is evident that the methodology is not adequately justified. Critical components, such as consequent measurement interactions, such as between hand and back laterality judgment tasks, or conducting maximal effort tests before motor imagery ones. Furthermore, the sample size is not enough for determining conclusive results, that summed to the problem with measurement procedures may lead to biased findings.

Despite the manuscript clearly is an interesting topic of research, their objectives are not clear. From my point of view, it is not suitable aiming to analyse such different issues, exploring the differences between LBP and healthy populations, but additionally aiming to determine the reliability of a test. They are completely different aims, which should be addressed with different sample sizes (usually ICC analyses requiere bigger samples than the amount included in the study), and addressed in a different order in the research line. First analysing and suitably of the test concluding its reliability, followed by the analysis of differences in that test between populations.".

Please revise the manuscript considering this feedback and reviewer 2 report to the authors.

Thank you.

Best regards.

Reviewer 1 ·

Basic reporting

nothing to report

Experimental design

nothing to report

Validity of the findings

nothing to report

Additional comments

nothing to report

Reviewer 2 ·

Basic reporting

Authors have adressed the concerns indicated in the previous review processes, stating the limitations of the study methodology in the Discussion section. Nevertheless, they were highly sugested to modify and re-conduct the recruitment and measurement procedures.

Experimental design

Authors are requested to provide the ICC data separately for healthy and LBP patients, as requested in the previous review.

Validity of the findings

Please, include in the methods section the use of Brown-Forsythe test. Please, additionally report the data of ICC separetedly for healthy and LBP patients.

---

## Round 0.4 · accepted · Accept

Dear Authors,

Thank you for addressing the reviewers' comments.

Best regards.